# Common Bacterial Infections during the 3-Month Period after SARS-CoV-2 Infection: A Retrospective Cohort Study

**DOI:** 10.3390/healthcare11243151

**Published:** 2023-12-12

**Authors:** Bar Cohen, Shirley Shapiro Ben David, Daniella Rahamim-Cohen, Afif Nakhleh, Arnon Shahar, Ilan Yehoshua, Avital Bilitzky-Kopit, Joseph Azuri, Miri Mizrahi Reuveni, Limor Adler

**Affiliations:** 1Health Division, Maccabi Healthcare Services, Tel Aviv 6812509, Israel; cobar@post.bgu.ac.il (B.C.); shapira_sr@mac.org.il (S.S.B.D.); cohen_dani@mac.org.il (D.R.-C.); shahar_a@mac.org.il (A.S.); yehoshua_i@mac.org.il (I.Y.); avitalbilitzky@gmail.com (A.B.-K.); azuri_yo@mac.org.il (J.A.);; 2The Faculty of Health Science, Ben Gurion University, Beer Sheva 8443944, Israel; 3Department of Family Medicine, Faculty of Medicine, Tel Aviv University, Tel Aviv 6997801, Israel; 4Diabetes and Endocrinology Clinic, Maccabi Healthcare Services, Haifa 3299001, Israel; nakhleh_a@mac.org.il; 5Institute of Endocrinology, Diabetes and Metabolism, Rambam Health Care Campus, Haifa 3109601, Israel; 6The Azrieli Faculty of Medicine, Bar-Ilan University, Safed 1311502, Israel

**Keywords:** bacterial infections, COVID-19, SARS-CoV-2, pneumonia, urinary tract infections, group A streptococcus pharyngitis

## Abstract

Introduction: Correlations between SARS-CoV-2 and bacterial infections have mainly been studied in hospitals, and these studies have shown that such interactions may be lethal for many. In the context of community flora, less is known of the trends and consequences of viral infections relative to subsequent bacterial infections. Purpose: This study aims to explore the prevalence and characteristics of bacterial infections in the three months following SARS-CoV-2 infections, in a community, real-world setting. Methods: In this retrospective cohort study, we compared patients who completed a polymerase chain reaction (PCR) test or an antigen test for SARS-CoV-2 during January 2022, the peak of the Omicron wave, and examined bacterial infections following the test. We searched these cases for diagnoses of the following four bacterial infections for three months following the test: Group A Streptococcus (GAS) pharyngitis, pneumonia, cellulitis, and urinary tract infections (UTI). Results: During January 2022, 267,931 patients tested positive and 261,909 tested negative for SARS-CoV-2. Test-positive compared to test-negative patients were significantly younger (42.5 years old vs. 48.5 years old, *p* < 0.001), smoked less, and had fewer comorbidities (including ischemic heart disease, diabetes mellitus, hypertension, chronic obstructive pulmonary disease, and chronic renal failure). In the multivariable analysis, test-positive patients had an increased risk for GAS pharyngitis (adjusted odds ratio [aOR] = 1.25, 95% CI 1.14–1.38, *p*-value < 0.001) and pneumonia (aOR = 1.25, 95% CI 1.15–1.35, *p*-value < 0.001), a trend towards an increased prevalence of UTI (aOR = 1.05, 95% CI 0.99–1.12, *p*-value = 0.092), and lower risk for cellulitis (aOR = 0.92, 95% CI 0.86–0.99, *p*-value < 0.05). Conclusions: A history of SARS-CoV-2 infection in the past three months increased susceptibility to respiratory tract bacterial infections and the prevalence of UTI.

## 1. Introduction

Severe acute respiratory syndrome coronavirus 2 (SARS-CoV-2) emerged in 2019 and was later declared by the World Health Organization to be a global pandemic in March 2020 [1]. The virus spread quickly, burdening healthcare systems worldwide, causing millions of deaths, and demonstrating various clinical presentations and complications [2]. While much attention was given to hospital care and severe cases, most patients were treated in a community setting and had only mild or asymptomatic infection [3,4].

Correlations between viral and bacterial infections have been extensively studied, especially in the context of pandemics [5,6]; influenza, and especially the H1N1 strain, has been one example where the viral infection often preceded a secondary and far more lethal bacterial infection. Moreover, studies found that in children, bacterial pneumonia is often accompanied by a concurrent or preceding viral pathogenic presence [7]. Mechanisms for post-viral bacterial infection include alteration of immune response, cell and tissue damage, microbial dysbiosis, and more [5,8,9]. In otherwise-healthy individuals, the post-viral bacterial infection is usually localized to the same organ or organ system. This appears to intuitively support the importance of tissue damage in the pathogenesis of post-viral bacterial infections. In the case of the respiratory system, specialized immune components and barriers take time to regenerate and normalize after damage, which may present bacterial infections with an opportunity.

Previous studies have investigated bacterial infections following SARS-CoV-2 [8]; however, most focused on hospitalized patients; thereby, such research efforts were focused on cases that were more severe and patients who were more vulnerable. Studies regarding infections in the post-acute period following infection with SARS-CoV-2 in a community, and in a real-world setting, are lacking. The distinction between the two environments is key due to both specific characteristics of patients—hospitalized patients are more vulnerable and more symptomatic—and the nosocomial flora, which is unique and can be highly pathogenic. Accordingly, the community setting allows us to investigate the natural history of the disease in a relatively healthy population in which both the pathophysiology and epidemiology of post-COVID19 bacterial infections can be demonstrated.

However, the effects of the pandemic restrictions in many countries confounded the relationships between SARS-CoV-2 infections and other viral and bacterial infections. Such restrictions included social distancing and remote daily activities, and dramatically lowered the exposure to other pathogens [10,11]. Therefore, the full extent of potential infections following a SARS-CoV-2 infection may be hard to identify. Previous studies showed lower levels of non-COVID-19 infections in younger populations such as children, who were highly susceptible and affected by the common restrictions [12]. In Israel, where this study takes place, by January 2022, most restrictions were lifted, and most people attended educational and occupational institutions regularly. While mask wearing was mandatory in closed spaces, all activities were routinely held. The COVID-19 morbidity associated with this time was mostly in relatively young, healthy patients. These features make this timeframe well-suited for looking into the community setting among healthy individuals. Additionally, this timeframe was mostly associated with the Omicron variant, which was less pathogenic than its counterparts, thus allowing us to identify the possible outcomes, even if tissue damage was not manifested symptomatically.

This study aimed to explore the prevalence and characteristics of common bacterial infections in a community setting, in January 2022, within the three months following a SARS-CoV-2 infection.

## 2. Methods

### 2.1. Study Design and Setting

In this retrospective nationwide cohort study, we evaluated all patients who tested positive or negative in a polymerase chain reaction (PCR) test or antigen test for SARS-CoV-2 in January 2022 with Maccabi Healthcare Services (MHS). MHS is the second-largest healthcare maintenance organization in Israel, and covers more than 2.6 million patients nationwide. Our cohort represents patients diagnosed with SARS-CoV-2 for the first time during the fifth wave of the pandemic in Israel, which was dominated by the Omicron variant. We followed all test-positive and test-negative patients for three months following the test. We evaluated the prevalence of four common bacterial infections: group A streptococcal (GAS) pharyngitis, urinary tract infection (UTI), pneumonia, and cellulitis. We chose these infections due to their common presentation in community care. It is worth mentioning that each of these infections is not uniquely caused by bacterial pathogens, but can also be caused by viruses and other pathogens. We compared the prevalence of these four infections between test-positive and test-negative patients and examined whether the prevalence had changed in the three months following the SARS-CoV-2 infection. Patients who tested negative in January, but who were later infected with a SARS-CoV-2 infection were excluded from this study, as were individuals who contracted SARS-CoV-2 previously. Additionally, we collected and controlled for variables known to be associated with infectious disease outcomes and prevalence; these are comorbidities, age, gender, ethnic background, and SES. SES was based on a national governmental average by location, and the 10 ordinal scores were divided into groups (low: 1–4, medium: 5–7, and high 8–10). All MHS members meeting the inclusion criteria were recorded in this study, making it a representative sample and allowing for a thorough investigation into the local and general effects of infection with SARS-CoV-2 on bacterial infections in the immediately subsequent timeframe.

### 2.2. Variables

We collected demographic and medical variables for all patients, including age, sex, ethnic background (i.e., Orthodox Jews, Arabs, all others), socioeconomic status (SES, with 1 being the lowest and 10 the highest), smoking status, and the presence of common chronic illnesses (including diabetes mellitus [DM], hypertension [HTN], ischemic heart disease [IHD], chronic obstructive pulmonary disease [COPD], chronic kidney disease [CKD], inflammatory bowel disease [IBD], oncologic disease, dementia, and osteoporosis). All of the abovementioned variables are documented automatically in the electronic medical records. No human intervention was needed for the recording of these variables. Therefore, missing data is rarely observed.

The outcome variables include four possible outcomes: (1) pharyngitis with a confirmed positive throat culture for GAS; (2) diagnosis of UTI, along with the fulfillment of prescription for antibiotics within a week of the diagnosis; (3) diagnosis of pneumonia, along with the fulfillment of a prescription for antibiotics within a week of the diagnosis; (4) diagnosis of cellulitis, plus fulfillment of prescription for antibiotics within a week of the diagnosis. Since long-COVID symptoms include throat pain and abdominal pain, we also collected the number of urine and throat cultures to eliminate this possible bias [13].

### 2.3. Statistical Analysis

Descriptive statistics were produced for all variables, with absolute numbers and percentages for categorical variables, and mean and standard deviation for continuous variables. We performed chi-square tests for categorical variables and t-tests for continuous variables to compare test-positive and test-negative patients. We then performed univariate analyses to compare the prevalence of the outcome variables in both groups. For each outcome, we performed a multivariable analysis using a logistic regression analysis with two blocks, the first using the ENTER approach for baseline characteristics, and the second the FORWARD approach for all other variables. The ENTER approach inserts all the variables into the model, whereas the FORWARD approach inserts only the significant variables into the model. For the multivariable analysis, we assessed the goodness-of-fit of the model with the Hosmer–Lemeshow method. In addition, to assess whether multicollinearity exists, we used the variance inflation factor (VIF). We used the Statistical Package for Social Sciences (SPSS) software, version 27, for data analysis.

## 3. Results

### 3.1. Participants

In January 2022, during the peak of the SARS-CoV-2 Omicron-variant wave in Israel, 267,931 people in MHS tested positive for SARS-CoV-2 infection (PCR or antigen test), and 261,909 tested negative. Test-positive and test-negative patients varied in demographics and as to comorbid diseases (Table 1). The positive and negative groups were distinct from one another demographically. In the test-positive group, there were more females (*p*-value < 0.05), more Ultra-Orthodox and Arab patients (*p*-value < 0.001), fewer high SES patients (*p*-value < 0.001), more non-smokers (*p*-value < 0.001), and fewer patients with chronic comorbidities (across all comorbidities checked, *p*-value < 0.001). These descriptive statistics emphasize that those who contracted SARS-CoV-2 during this time were relatively healthy individuals. This also demonstrates that the test-negative group is statistically more vulnerable to the bacterial infections tested, due to the higher rates of comorbidities. The full descriptive and basic comparative statistics for the sample are presented in Table 1.

### 3.2. Univariate Analysis

During the three months following the SARS-CoV-2 test, GAS pharyngitis was more prevalent in test-positive patients (1741 [0.65%] vs. 1176 [0.45%], *p* < 0.001). Conversely, UTI, cellulitis, and pneumonia were more prevalent in test-negative patients (2939 [1.12%] vs. 2767 [1.03%], *p*-value =0.002; 1963 [0.75%] vs. 1561 [0.58%], *p*-value <0.001; and 1420 [0.54%] vs. 1324 [0.49%], *p*-value <0.001, respectively). Therefore, the univariate analysis suggests a mixed trend, where GAS is distinctly associated with the post-viral state. The full prevalence of bacterial infections after SARS-CoV-2 infection is presented in Figure 1.

### 3.3. Multivariable Analyses

Infection with SARS-CoV-2 was associated with higher odds for GAS pharyngitis and pneumonia (aOR = 1.25, 95% CI 1.14–1.38, *p*-value < 0.001; and aOR = 1.25, 95% CI 1.15–1.35, *p*-value <0.001, respectively) (Table 2 and Table 3). There was a trend towards an increase in the prevalence of UTI among patients with SARS-CoV-2 infection (aOR = 1.05, 95% CI 0.99–1.12, *p*-value =0.092) (Table 4). However, cellulitis was less prevalent in patients with a history of SARS-CoV-2 infection (aOR = 0.92, 95% CI 0.86–0.99, *p*-value < 0.05) (Table 5). In the case of pneumonia, additional factors appeared to be dominant, and directly affected the odds, including female gender, smoking in the past, and comorbidities (with a *p*-value < 0.001). For pneumonia, individuals with high SES and high COVID-19 vaccination status have lower odds of contracting the bacterial infection. For UTI, as expected, females are at higher risk of disease (*p*-value < 0.001), and all listed comorbidities but IBD demonstrate a direct correlation with the infection (*p*-value < 0.001). Interestingly, COVID-19 vaccination status is also directly correlated with greater risk of UTI. Finally, additional factors directly affecting the odds of cellulitis are female gender (*p*-value = 0.005), Ultra-Orthodox ethnic background (*p*-value = 0.002), smoking status (*p*-value < 0.001) and comorbidities (for all, *p*-values < 0.001). While these comorbidities and smoking status are known to affect vascular factors important for immune response and wound healing, gender and ethnic background correlations need to be better investigated. Individuals reporting having high or medium SES had reduced odds of developing cellulitis, compared to those with low SES (*p*-value < 0.001).

## 4. Discussion

### 4.1. Main Findings

This study compared the prevalence of four common bacterial infections in a community setting during the three months following a SARS-CoV-2 test. We report an increased risk for GAS pharyngitis and pneumonia (25% increase each) and a lower risk for cellulitis (8% risk reduction). We also identify a trend towards an increase in the prevalence of UTI (5% increase).

### 4.2. Interpretation

We have shown increases in respiratory system bacterial infections (GAS pharyngitis and pneumonia) in the three months following infection with SARS-CoV-2. Manna et al. suggest that viral infections damage respiratory airways and both the innate and acquired immune responses, thus providing a favorable environment for bacterial infections in the respiratory tract [8]. Further establishing this correlation are the decreased risks for pneumonia and GAS pharyngitis in individuals vaccinated against COVID-19. For pneumonia, the risk for the infection consistently decreases for any number of vaccines (1–2 vaccines—OR = 0.79, *p*-value < 0.05; 3–4 vaccines—OR = 0.6, *p*-value < 0.001).

There are known pulmonary sequela after infection with SARS-CoV2 [14]. Interestingly, the univariate analysis showed that pneumonia was more prevalent in test-negative patients. However, this association was reversed in the multivariable analysis (when pneumonia was more prevalent among test-positive patients). This can be explained by the differences seen between test-negative and test-positive patients. In the Omicron wave, patients were younger, and had fewer comorbidities. As seen in Table 1, test-negative patients showed more IHD, DM, HTN, and CKD than did test-positive patients, and fewer were smokers. Thus, when performing a univariate analysis, it was reasonable to find a higher prevalence of pneumonia in this population. However, the outcome was reversed when eliminating these factors in the multivariable analysis.

We also found that women are at increased risk for all the infections tested (GAS (OR = 1.11, *p*-value <0.05), UTI (OR = 6.20, *p*-value <0.001), cellulitis (OR = 1.1, *p*-value < 0.05), pneumonia (OR = 1.2, *p*-value <0.001)). There are gender-based differences in the rates of bacterial infections [15]. These differences are related to both biologic sex and gender roles (when mothers are typically the primary caregivers for children). UTIs are more common among women, due to several reasons, including the shorter female urethra and the proximity between the rectum and vagina [15,16,17]. While upper respiratory tract infections are more common in women, lower respiratory tract infections, including pneumonia, are normally more prevalent in men [16,18,19]. Regarding cellulitis, there is inconsistent evidence about gender-related differences. Some point to an absence of difference [20], while others suggest that it predominantly affects females [21].

Increased risks for pneumonia and cellulitis were also found to be associated with an Ultra-Orthodox ethnic background (OR = 1.25, *p* = 0.007; OR = 1.27, *p*-value < 0.05, respectively). This may be related to special characteristics of this subgroup, including having more children compared to the secular population, living in conditions which are more crowded, and having lower rates of vaccination [22,23].

The risks for cellulitis and pneumonia were increased for patients with HTN, DM, CKD, cardiovascular disease, and cancer. This finding is consistent with other studies [6,24,25]. The risk for UTI was increased among patients with diabetes and cardiovascular disease and decreased for patients with IBD. Comorbidities such as these might reduce immunological abilities working to fight infections and lead to environmental conditions that are hospitable to infections [26]. In addition, medications used to treat DM, such as SGLT-2 inhibitors and others, are associated with an increased risk for UTI, especially in high doses [27,28]. Moreover, the risk for pneumonia was increased for COPD patients (OR = 2.89, *p*-value <0.001), a finding which is consistent with the existing research [25]. The risk for pneumonia was decreased in the high SES group (OR = 0.75, *p*-value < 0.001) [25], as were the risks for cellulitis (OR = 0.71, *p*-value < 0.001) [29] and UTI (OR = 0.87, *p*-value < 0.05) [30]. Only in the case of cellulitis is this phenomenon also reported in the intermediate SES group (OR = 0.83, *p*-value < 0.001). Further studies should investigate these differences.

History, including very recent history, has been shaped by pandemics and their effects on societies. The field of public health originated in the efforts to protect communities from pathogens, long before those pathogens were even recognized. Epidemiology and trends detected in pandemics expand our knowledge and help us better treat future patients. Our study echoes lessons from the Spanish flu, after which many died from subsequent bacterial infections. We were able to recognize risk factors and the localized effect of the virus on the respiratory system. This can help healthcare systems to target vulnerable patients, focus their efforts on preventative medicine, and better equip physicians and clinics for the aftermath of viral infections.

## 5. Strengths and Limitations

This study is unique in its focus on community-acquired infections following SARS-CoV-2 and is based on a representative nationwide sample suitable for this purpose. We used a large cohort, with all relevant sociodemographic and medical variables available in the medical records.

However, this study has several limitations. First, while this large sample size is a strength, it also brings about significant correlations that may result from sheer volume. Additionally, three of the four infections we investigated (UTI, pneumonia, and cellulitis) were clinically diagnosed. This means that one cannot confirm the bacterial etiology of these diagnoses. However, given the diagnostic guidelines and the common causes of these infections, we suggest that they are most likely attributable to bacteria. Pneumonia is worthy of particular notice, since often, viral infections may be mislabeled as bacterial pneumonia; therefore, this might be an even less accurate diagnosis when based on the data. Additionally, the two samples may also vary due to their characteristics. The test-negative sample includes more patients with chronic illnesses, and who may be more inclined to be tested once even a minor symptom occurs, while healthier patients may choose to be tested only in the presence of more serious symptoms. While we only tested for SARS-CoV-2, it should be noted that there is always a possibility for another viral infection we may not be aware of. This is especially significant for influenza, which is also known to correlate with bacterial infections. Moreover, this study is based on data for the Omicron variant, and we cannot assume that all variants have exactly the same characteristics; therefore, the findings may not be directly applicable to other variants.

## 6. Conclusions

Understanding the nuanced interactions between viruses and bacteria may improve the care and treatment of patients. We report a statistically significant association between a history of SARS-CoV-2 infection and bacterial infections in the respiratory tract in the following three months. This effect was limited to the respiratory system, which was previously targeted by the viral infection. We also suggest a trend towards a higher prevalence of UTI and lower odds of cellulitis. Additionally, we were able to establish the effects of SES and SARS-CoV-2 vaccination status on bacterial infections after the viral infection. Such insights enable us to recognize patients at higher risk of morbidity and assess potential collateral features of viral disease.

## Figures and Tables

**Figure 1 healthcare-11-03151-f001:**
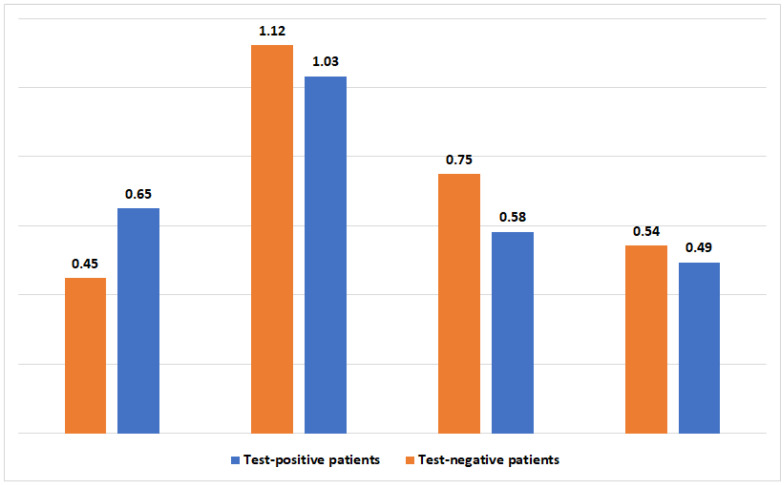
The prevalence (in percentages) of four bacterial infections 3 months following a positive or negative SARS-CoV-2 test.

**Table 1 healthcare-11-03151-t001:** Characteristics of test-negative and test-positive patients during January 2022.

Variables	Test-Positive Patients (*n* = 267,931)	Test-Negative Patients (*n* = 261,909)	*p*-Value
Gender			
Men, % (*n*)	42.8% (114,725)	45.0% (117,842)	
Women, % (*n*)	57.2% (153,206)	55.0% (144,067)	0.001
Ethnic background			
Ultra-Orthodox, % (*n*)	6.1% (16,474)	3.8% (9843)	
Arabs, % (*n*)	5.2% (13,833)	4.5% (11,826)	
Other (*n*)	88.7% (237,612)	91.7% (240,221)	<0.001
Socioeconomic status,			
Low (1–4), % (*n*)	16.6% (44,349)	15.1% (39,553)	
Intermediate (5–6), % (*n*)	31.9% (85,382)	30.1% (78,760)	
High (7–10), % (*n*)	51.6% (138,188)	54.8% (143,577)	<0.001
Smoking			
Non-smoker, % (*n*)	84.3% (225,979)	80.8% (211,544)	
Current smoker, % (*n*)	10.6% (28,521)	14.4% (37,625)	
Past smoker, % (*n*)	1.4% (3871)	1.4% (3779)	
Unknown, % (*n*)	3.6% (9560)	3.4% (8961)	<0.001
Age (mean ± SD)	42.5 ± 15.6	48.5 ± 18.2	<0.001
Vaccination status			
Unvaccinated, % (*n*)	10.3% (27,472)	6.0% (15,638)	<0.001
1–2 vaccines, % (*n*)	15.3% (40,975)	10.9% (28,547)	
3–4 vaccines, % (*n*)	74.5% (199,484)	83.1% (217,724)	
Cardiovascular disease, % (*n*)	3.4% (9213)	6.6% (17,344)	<0.001
Diabetes, % (*n*)	6.3% (16,800)	10.3% (27,028)	<0.001
Hypertension, % (*n*)	14.0% (37,510)	22.6% (59,234)	<0.001
COPD, % (*n*)	1.0% (2736)	2.3% (6056)	<0.001
Chronic kidney disease, % (*n*)	3.1% (8355)	6.3% (16,581)	<0.001
Cancer diagnosis, % (*n*)	5.1% (13,550)	9.0% (23,657)	<0.001

**Table 2 healthcare-11-03151-t002:** Variables associated with streptococcal pharyngitis; multivariable analysis.

Variables	Adjusted OR (95%CI)	*p*-Value
A history of infection with SARS-CoV-2 in the previous three months	1.25 (1.14–1.38)	<0.001
Women	1.11 (1.00–1.22)	0.046
Ethnic background		
Ultra-Orthodox	1.18 (0.97–1.44)	0.09
Arabs	0.77 (0.56–1.04)	0.08
Other	Reference	
Socioeconomic status		
Low (1–4)	Reference	
Intermediate (5–6)	1.01 (0.86–1.20)	0.86
High (7–10)	1.04 (0.87–1.23)	0.68
Smoking		
Non-smoker	Reference	
Current smoker	0.87 (0.74–1.02)	0.09
Past smoker	1.12 (0.77–1.62)	0.56
Unknown	0.83 (0.62–1.11)	0.22
Age	0.99 (0.983–0.989)	<0.001
Vaccination		
Unvaccinated	Reference	
1–2 vaccines	0.83 (0.68–1.02)	0.07
3–4 vaccines	0.90 (0.76–1.08)	0.26
Number of throat cultures	413.7 (365.5–468.4)	<0.001

**Table 3 healthcare-11-03151-t003:** A multivariable logistic regression model evaluating associations with pneumonia.

Variables	Adjusted OR (95%CI)	*p*-Value
A history of infection with SARS-CoV-2 in the previous three months	1.25 (1.15–1.35)	<0.001
Women	1.20 (1.11–1.30)	<0.001
Ethnic background		
Ultra-Orthodox	1.25 (1.06–1.48)	0.007
Arabs	1.19 (0.99–1.43)	0.06
Other	Reference	
Socioeconomic status		
Low (1–4)	Reference	
Intermediate (5–6)	1.00 (0.89–1.12)	0.97
High (7–10)	0.75 (0.67–0.85)	<0.001
Smoking		
Non-smoker	Reference	
Current smoker	0.94 (0.83–1.06)	0.33
Past smoker	1.39 (1.07–1.81)	0.01
Unknown	0.42 (0.27–0.63)	<0.001
Age	1.032 (1.029–1.035)	<0.001
Vaccination		
Unvaccinated	Reference	
1–2 vaccines	0.79 (0.67–0.92)	0.003
3–4 vaccines	0.60 (0.53–0.68)	<0.001
Cardiovascular disease	1.60 (1.43–1.79)	<0.001
Diabetes mellitus	1.23 (1.11–1.36)	<0.001
Hypertension	1.27 (1.15–1.40)	<0.001
COPD	2.89 (2.54–3.29)	<0.001
Chronic kidney disease	1.21 (1.07–1.36)	0.001
Cancer diagnosis	1.38 (1.24–1.53)	<0.001

**Table 4 healthcare-11-03151-t004:** A multivariable logistic regression model evaluating associations with urinary tract infection.

Variables	Adjusted OR (95%CI)	*p*-Value
A history of infection with SARS-CoV-2 in the previous three months	1.05 (0.99–1.12)	0.092
Women	6.20 (5.64–6.81)	<0.001
Ethnic background		
Ultra-Orthodox	0.81 (0.71–0.95)	0.008
Arabs	1.04 (0.90–1.21)	0.563
Other	Reference	
Socioeconomic status		
Low (1–4)	Reference	
Intermediate (5–6)	0.93 (0.85–1.03)	0.165
High (7–10)	0.87 (0.79–0.96)	0.006
Smoking		
Non-smoker	Reference	
Current smoker	1.13 (1.03–1.24)	0.01
Past smoker	0.91 (0.70–1.19)	0.502
Unknown	0.64 (0.48–0.84)	0.001
Age	1.014 (1.011–1.016)	<0.001
Vaccination		
Unvaccinated	Reference	
1–2 vaccines	1.23 (1.08–1.41)	0.002
3–4 vaccines	1.21 (1.07–1.35)	0.002
Number of urine cultures	5.23 (5.08–5.38)	<0.001
Cardiovascular disease	1.27 (1.13–1.44)	<0.001
Hypertension	1.21 (1.12–1.32)	<0.001
IBD	0.60 (0.43–0.82)	0.002

**Table 5 healthcare-11-03151-t005:** A multivariable logistic regression model evaluating associations with cellulitis.

Variables	Adjusted OR (95%CI)	*p*-Value
A history of infection with SARS-CoV-2 in the previous three months	0.92 (0.86–0.99)	0.02
Women	1.10 (1.03–1.18)	0.005
Ethnic background		
Ultra-Orthodox	1.27 (1.09–1.46)	0.002
Arabs	1.03 (0.88–1.21)	0.7
Other	Reference	
Socioeconomic status		
Low (1–4)	Reference	
Intermediate (5–6)	0.83 (0.75–0.92)	<0.001
High (7–10)	0.71 (0.64–0.78)	<0.001
Smoking		
Non-smoker	Reference	
Current smoker	1.25 (1.13–1.37)	<0.001
Past smoker	1.61 (1.28–2.01)	<0.001
Unknown	0.51 (0.38–0.69)	<0.001
Age	1.012 (1.010–1.015)	<0.001
Vaccination		
Unvaccinated	Reference	
1–2 vaccines	1.05 (0.90–1.22)	0.53
3–4 vaccines	0.99 (0.87–1.13)	0.89
Cardiovascular disease	1.41 (1.26–1.58)	<0.001
Diabetes mellitus	1.35 (1.22–1.49)	<0.001
Hypertension	1.41 (1.29–1.55)	<0.001
Chronic kidney disease	1.29 (1.15–1.45)	<0.001
Cancer diagnosis	1.23 (1.10–1.36)	<0.001

## Data Availability

The data supporting this study are available from the corresponding author, but restrictions apply to the availability of this information, which were used under license for the current study, and so are not publicly available. Data are, however, available from the authors upon reasonable request and with permission of the local ethics committee of MHS.

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
