# Peer review of "Common Bacterial Infections during the 3-Month Period after SARS-CoV-2 Infection: A Retrospective Cohort Study"

_healthcare, 2023, doi:10.3390/healthcare11243151_

Round 1

Reviewer 1 Report

Comments and Suggestions for Authors

Thank you for the efforts done in this manuscript.

Actually this study is a misleading and add nothing to the literature, due to several reasons:

1. The two groups (positive versus negative) are significantly different, making the comparison and conclusion not valid in the absence of a true control/reference group.

2. The results section contains several weakness. Although there are many statisticall significant findings, they are not helpful due to the previous comment and

3. The OR of 1.25 means weak association, the authors overestimate the association in the interpretation.

4. The title and text concentrate mainly on bacterial infection, however, no true bacterial detection was made, for example, pneumonia can be caused by viruses; depending on the studies the viral (other than SARS-CoV2)-associated pneumonia is increasing after the pandemic (more than 25%). This make the whole study misleading. Mention that in the limitations is not enough.

5. What is the rational of the link between COVID-19 and cellulitis or UTI? I can not find any justification mentioned in the manuscript.

6. The first half part of the interpretaion in the discussion (the authors did not use line numbers to refer) contain several drawback in the writing and interpretation with caution.

7. The authors did not compare well with other related articles worldwide and in the region of the study (Middle East).

8. There is no mention of the co-infection, I mean there are several studies related to the co-infection between virus-bacteria and virus-fungus like Pneumocystis with a confirmed diagnosis (not depending only on clinical features). Those studies are better and stronger than what mentioned in this manuscript.

Minor comment:

Some parts in the introduction can be moved to the methods.

Comments on the Quality of English Language

Good

Author Response

Thank you for the efforts done in this manuscript.

Actually this study is a misleading and add nothing to the literature, due to several reasons:

  1. The two groups (positive versus negative) are significantly different, making the comparison and conclusion not valid in the absence of a true control/reference group.

ANSWER: As in many other COVID studies, we believe it is reasonable to compare test-positive and test-negative patients. The multivariate analysis addresses the differences seen in Table 1.

  1. The results section contains several weaknesses. Although there are many statistical significant findings, they are not helpful due to the previous comment and
  2. The OR of 1.25 means weak association, the authors overestimate the association in the interpretation.

ANSWER: It is a weak but significant association.

  1. The title and text concentrate mainly on bacterial infection, however, no true bacterial detection was made, for example, pneumonia can be caused by viruses; depending on the studies the viral (other than SARS-CoV2)-associated pneumonia is increasing after the pandemic (more than 25%). This make the whole study misleading. Mention that in the limitations is not enough.

ANSWER:

We added this to the study design:

We chose these infections due to their common presentation in community care. It is worth mentioning that all these infections are not merely caused by bacterial pathogens, but can also be caused by viruses and other pathogens.  

  1. What is the rational of the link between COVID-19 and cellulitis or UTI? I can not find any justification mentioned in the manuscript.

ANSWER: We were looking for common bacterial infections in the community.

  1. The first half part of the interpretaion in the discussion (the authors did not use line numbers to refer) contain several drawback in the writing and interpretation with caution.

ANSWER: We made substantial changes to the discussion, and we believe it is improved.

  1. The authors did not compare well with other related articles worldwide and in the region of the study (Middle East).

ANSWER: We elaborated on the discussion and added relevant articles worldwide and regional.

  1. There is no mention of the co-infection, I mean there are several studies related to the co-infection between virus-bacteria and virus-fungus like Pneumocystis with a confirmed diagnosis (not depending only on clinical features). Those studies are better and stronger than what mentioned in this manuscript.

Minor comment:

Some parts in the introduction can be moved to the methods.

Reviewer 2 Report

Comments and Suggestions for Authors

I do not think this piece adds anything. It is well known that GAS and S.aureus pnuemonia can figure  There are some severe limitations particularly the clinical diagnosis of the 4 major infections with no microbiological analysis (outside of the throat swab). We do not discuss the timing of these infections within the 3 months and whether patients had pre-existing UTI diagnoses or hospitalisations for other reasons during this time . There is no valuable radiological or microbiological data regarding pneumonia and UTI. 

We also do not discuss underlying immunosuppression and the severity of the underlying COVID infection.  How many of the individuals had children with diagnosed GAS ? etc.

Author Response

Thank you for your feedback. We have made substantial changes in the manuscript and hope it is improved. 

Reviewer 3 Report

Comments and Suggestions for Authors

Thank you for your work in this area. Your paper uses a rich dataset to report the odds of infection following testing positive for COVID-19. I did feel some areas needed more work, such as: adding more details to the methods section on the dataset, data cleaning process, and model fitting/post-estimation diagnostics. Moreover, I also noticed some confusion in terminology (e.g., univariate, multivariate, and correlations) being used. Lastly, and perhaps most importantly, I felt the discussion and conclusion needed to be reworked and expanded greatly. I would advise the authors to carefully review my comments below before revising and resubmitting it.

1. Introduction, 1st paragraph. "The virus spread quickly, causing significantly burdening healthcare systems..." reads strange. I would suggest deleting "causing significantly" so it reads "...spread quickly, burdening healthcare systems..."

2. Methods. Since you seem to be using an administrative health database linked to MHS which allowed for this retrospective study, can you provide details on: (i) missing data and how you dealt with that? (ii) which variables you recoded for analysis (e.g., was SES already in 3 categories in the original database or did you collapse it into 3 categories for regression?)

3. Methods, Section 2.3. "We performed a univariate analysis using Chi-Square tests" change to "We used Chi-square test..." Univariate analyses implies descriptive statistics with one variable.

4. Methods, Section 2.3. Change "we performed a multivariate analysis using a logistic regression analysis" to "we performed a multivariable analysis using..." Accordingly, please change "multivariate" to "multivariable" throughout your paper.

5. Methods, Section 2.3. Could you please describe the "ENTER" and 'FORWARD" approach here?

6. Methods, Section 2.3. Did you not have assess goodness-of-fit, multicollinearity, and other assumptions of logistic regression? If not, I would strongly suggest you do or state it somewhere in your limitations that it wasn't done.

7. Figure 1. Could you please change the text and numbers to black font? It is hard to see when using the default Microsoft Word/Excel grey font.

8. Results.  The tables are fine, but I see some issues in reporting in the text. For example, "smoking in the past, and comorbidities (with a p-value 

=<0.001)." What is =<? Did you mean to say "p ≤ 0.001" or "p < 0.001"? Please fix as I see this in other places of your submission too. The standard practice is to report "p < 0.05" in the text rather than "p = 0.046", and "p < 0.001" rather than "p =< 0.001".

9. Results. "For cellulitis, high and medium SES demonstrate inverse correlations" Rather than using the word "correlations", I would strongly suggest sticking closely with "odds" because we don't want to confuse the reader into thinking correlation coefficients were computed. In this case, I would change the wording to "Individuals reporting having high and medium SES had reduced odds of developing cellulitis compared to those with low SES." Please try to use similar wording in your other sentences under section 3.3.

10. Tables 3-5. Could you please update the table captions to be more detailed rather than just "Variables associated with cellulitis, multivariate analysis"? For example, "A multivariable logistic regression model evaluating associations with cellulitis among Israelis (n = ####)".

11. Discussion. Your discussion is a little weak and could use more substance. I would suggest adding a paragraph to answer questions such as: what does this means for future public health interventions and crises, are there any infections in particular that surveillance should focus on based on your findings, and what can be done to mitigate this? Then, cite an additional 5-10 articles. It looks rushed if you have < 25 references because you're not building on previously published literature.

12. Limitations. It would be worth adding here that this study was conducted using data from the Omicron variant period, which may affect external validity. Thus, observational findings from other periods with new variants/subvariants may show different results.

12. Conclusion. Change "We report a significant correlation..." to "We report a statistically significant association between". I also think you need to expand your conclusion section. I would suggest listing the 2-3 key findings (e.g., women having greater odds of acquiring other infections after testing positive for COVID-19 compared to men), and close with 1-2 sentences on areas for further investigation. Aim for 150-200 words for this section.

Comments on the Quality of English Language

I have included my language-specific feedback in my main comments.

Author Response

  1. Introduction, 1st paragraph. "The virus spread quickly, causing significantly burdening healthcare systems..." reads strange. I would suggest deleting "causing significantly" so it reads "...spread quickly, burdening healthcare systems..."

ANSWER: We have changed the phrasing in accordance with this recommendation.

  1. Methods. Since you seem to be using an administrative health database linked to MHS which allowed for this retrospective study, can you provide details on: (i) missing data and how you dealt with that? (ii) which variables you recoded for analysis (e.g., was SES already in 3 categories in the original database or did you collapse it into 3 categories for regression?)

ANSWER: (ii) information was added about the variable SES, in the methods section.

Regarding missing data, we added: All the abovementioned variables are documented automatically in the electronic medical records. No human intervention is needed for these variables. Therefore, missing data is rarely observed.

  1. Methods, Section 2.3. "We performed a univariate analysis using Chi-Square tests" change to "We used Chi-square test..." Univariate analyses implies descriptive statistics with one variable.

ANSWER: We have changed the phrasing in accordance with this recommendation.

  1. Methods, Section 2.3. Change "we performed a multivariate analysis using a logistic regression analysis" to "we performed a multivariable analysis using..." Accordingly, please change "multivariate" to "multivariable" throughout your paper.

ANSWER: We have changed the terminology in accordance with this recommendation.

  1. Methods, Section 2.3. Could you please describe the "ENTER" and 'FORWARD" approach here?

ANSWER: We added the explanation to the text.

The ENTER approach inserts all the variables into the model, whereas the FORWARD approach inserts only the significant variables into the model.

  1. Methods, Section 2.3. Did you not have assess goodness-of-fit, multicollinearity, and other assumptions of logistic regression? If not, I would strongly suggest you do or state it somewhere in your limitations that it wasn't done.

ANSWER:

We added this to the text:

For the multivariate analysis, we assessed the goodness-of-fit of the model with the Hosmer-Lemeshow method. In addition, to assess whether multicollinearity exists, we used the variance inflation factor (VIF).

  1. Figure 1. Could you please change the text and numbers to black font? It is hard to see when using the default Microsoft Word/Excel grey font.

ANSWER: yes, we changed it to black and bold so it will be more easily read.

  1. Results.  The tables are fine, but I see some issues in reporting in the text. For example, "smoking in the past, and comorbidities (with a p-value 

=<0.001)." What is =<? Did you mean to say "p ≤ 0.001" or "p < 0.001"? Please fix as I see this in other places of your submission too. The standard practice is to report "p < 0.05" in the text rather than "p = 0.046", and "p < 0.001" rather than "p =< 0.001".

ANSWER: we corrected this throughout the text.

  1. Results. "For cellulitis, high and medium SES demonstrate inverse correlations" Rather than using the word "correlations", I would strongly suggest sticking closely with "odds" because we don't want to confuse the reader into thinking correlation coefficients were computed. In this case, I would change the wording to "Individuals reporting having high and medium SES had reduced odds of developing cellulitis compared to those with low SES." Please try to use similar wording in your other sentences under section 3.3.

ANSWER: we re-phrased and re-worded this section to better reflect our methods.

  1. Tables 3-5. Could you please update the table captions to be more detailed rather than just "Variables associated with cellulitis, multivariate analysis"? For example, "A multivariable logistic regression model evaluating associations with cellulitis among Israelis (n = ####)".

ANSWER: We have changed the phrasing in accordance with this recommendation.

  1. Discussion. Your discussion is a little weak and could use more substance. I would suggest adding a paragraph to answer questions such as: what does this means for future public health interventions and crises, are there any infections in particular that surveillance should focus on based on your findings, and what can be done to mitigate this? Then, cite an additional 5-10 articles. It looks rushed if you have < 25 references because you're not building on previously published literature.

ANSWER: we added to the discussion.

  1. Limitations. It would be worth adding here that this study was conducted using data from the Omicron variant period, which may affect external validity. Thus, observational findings from other periods with new variants/subvariants may show different results.

ANSWER: This additional and accurate limitation was added to the limitations and strengths section.

  1. Conclusion. Change "We report a significant correlation..." to "We report a statistically significant association between". I also think you need to expand your conclusion section. I would suggest listing the 2-3 key findings (e.g., women having greater odds of acquiring other infections after testing positive for COVID-19 compared to men), and close with 1-2 sentences on areas for further investigation. Aim for 150-200 words for this section.

ANSWER: We added and expanded this section.

Reviewer 4 Report

Comments and Suggestions for Authors

I consider that the idea of ​​this work is original and interesting, since the knowledge that could be generated would help understand the interactions between viruses and bacteria, and thereby achieve better therapeutic management of patients. However, as the authors mention, the work has very important limitations. My main criticism is that the absence of certainty in the etiology of the infection does not support the title or the conclusions of the work. I suggest not emphasizing bacterial infection but rather focusing the work on pharyngitis, UTI, pneumonia and cellulitis in general. I think it is convenient to change the title to: Common infections during the 3 months post SARSCoV-2 infection: a retrospective cohort study

 The discussion of the work should be more detailed, for example, the authors write:

"...We also found that women are at increased risk for all the infections tested …" But what is the possible explanation for this finding?

"…Increased risk for pneumonia and cellulitis were also found in the Ultra-Orthodox sector …" but what is the relevance of this?

Minor modifications

1. In the abstract, please separate introduction and aim.

2. Explain why you chose those four infections: pharyngitis, UTI, pneumonia, and cellulitis.

3. It would be interesting if they showed, in addition to the number of doses, the type of vaccine that the patients received

Author Response

 The discussion of the work should be more detailed, for example, the authors write:

"...We also found that women are at increased risk for all the infections tested …" But what is the possible explanation for this finding?

"…Increased risk for pneumonia and cellulitis were also found in the Ultra-Orthodox sector …" but what is the relevance of this?

ANSWER: 

We elaborated on the discussion about gender and ethnic differences.

Minor modifications

  1. In the abstract, please separate introduction and aim.

ANSWER: The abstract was re-phrased in accordance with this comment.

  1. Explain why you chose those four infections: pharyngitis, UTI, pneumonia, and cellulitis.

ANSWER: we added this to the METHODS section:

We chose these infections due to their common presentation in community care. It is worth mentioning that all these infections are not merely caused by bacterial pathogens, but can also be caused by viruses and other pathogens.

  1. It would be interesting if they showed, in addition to the number of doses, the type of vaccine that the patients received.

ANSWER: Unfortunately, we do not have this data in our study. But most vaccines in Israel were Pfizer's vaccines.

Round 2

Reviewer 1 Report

Comments and Suggestions for Authors

Most (if not all) of the comments are not addressed. One reason, because there are several critical issues that can not be resolved unless another new research is done.

Comments on the Quality of English Language

Moderate editing of English language required

Author Response

We addressed all comments of the reviewers as much as we could. 

Reviewer 2 Report

Comments and Suggestions for Authors

There has been some improvements. I do not think this demonstrates any new findings but has no objectionable content. 

Comments on the Quality of English Language

I do not think this really is a paradigm shifting piece of work. This revamps existing knowledge. I have suggested to accept the manuscript in its current form as it while it is not noteworthy it may be of some interest to readers and remind the need for surveillance of common infections post SARS-CoV2 infection. 

Author Response

Thank you for your suggestions. 

Reviewer 3 Report

Comments and Suggestions for Authors

Thank you for revising your manuscript and addressing my comments appropriately - that means a lot to me as a reviewer and I feel your paper is much stronger than before.

Comments on the Quality of English Language

None.

Author Response

Thank you. 

Reviewer 4 Report

Comments and Suggestions for Authors

Thanks for taking the suggestions.

Author Response

Thank you.